# Mechanical and Adsorptive Properties of Foamed EVA-Modified Polypropylene/Bamboo Charcoal Composites

**DOI:** 10.3390/ma14061524

**Published:** 2021-03-20

**Authors:** Wenzhu Li, Jing Zhang, Jingda Huang, Yuanchao Shao, Wenbiao Zhang, Chunping Dai

**Affiliations:** 1School of Engineering, Zhejiang A&F University, Hangzhou 311300, China; lwz@zafu.edu.cn (W.L.); zj12345662020@163.com (J.Z.); Shao19981211@163.com (Y.S.); 2Department of Wood Science, Faculty of Forestry, University of British Columbia, 2900-2424 Main Mall, Vancouver, BC V6T 1Z4, Canada; chunping.dai@ubc.ca

**Keywords:** bamboo charcoal, polypropylene, ethylene vinyl acetate, formaldehyde, foamed composite

## Abstract

Due to its excellent adsorption and humidity control function, bamboo charcoal (BC) has often been mixed with polypropylene (PP) to produce PP/BC composites for interior paneling applications. However, due to the poor foaming quality of PP, PP/BC composites suffer as a result of their high density, which limits their scope of use. Here, to improve its foaming quality, PP was modified with ethylene vinyl acetate (EVA), and then the EVA-modified PP (E-PP) was mixed with different contents of BC (0 wt.%–50 wt.%), as well as foaming agent (Azodicarbonamide, AC) and its auxiliaries (ZnO, Znst), in a twin-screw extruder, followed by hot-pressing at high temperature to obtain foamed E-PP/BC composites. The resulting composites showed good porosity and pore distribution with an increase of BC content by up to 20%. Further increase in the BC content seemed to cause the foaming performance to decrease significantly. The product density and adsorption rate increased, while the mechanical strength decreased with increasing BC content. At a BC content of 40 wt.%, the foamed E-PP/BC composite showed the best combined performance, with a density of 0.90 g/cm^3^, 24-h formaldehyde adsorption rate of 0.48, and bending strength of 11.59 MPa.

## 1. Introduction

Plastic materials have been developed and used widely for more than one hundred years because of their good processing property, chemical stability, electrical insulation, and impact resistance [1,2,3]. For the foreseeable future, plastics will remain an indispensable industrial material [4], despite their pollution problems [5,6]. In the meantime, efforts have been made to recycle plastics like metal materials [7,8], or to mix them with other fillers to reduce the white pollution [9,10].

Polypropylene (PP), as a common and low-density (only 0.89–0.91 g/cm^3^) plastic, has been well studied in terms of its substitution by more environmentally friendly fillers in order to achieve improvements in impact and aging resistance [11,12,13]. Studies have also been conducted to reduce PP density by foaming [14] and improving the properties of oil absorption and electric conduction [15,16]. However, PP is not as easy to foam as other plastics, such as polyurethane (PU), polystyrene (PS), and polyvinylchloride (PVC) [17]. Due to the inherent fluidity of PP, when the temperature exceeds its melting point, the melt viscosity drops rapidly. As a result, PP’s foaming temperature range is relatively narrow, close to its crystallization point [18,19,20].

Plastic-based composites have also been investigated as a way to reduce plastic usage and introduce new functions [21,22]. For example, wood–plastic composites [23] possess the durability of plastic and the biomass performance of wood; plastic–rubber composites [24] improve the rigidity of plastic; plastic–carbon fiber composites [25] modify the brittleness of plastic. Bamboo charcoal (BC) is a pyrolysis of solids from bamboo or bamboo residues with a highly porous structure that is suited for a wide variety of applications, including water purification, architectural decoration, daily chemicals, and the food industry [26,27]. BC is especially suited for the removal of formaldehyde, which is commonly found in interior wood composites [28,29,30,31]. It is an emerging green industrial material and is playing an important role in the development of the low-carbon and circular economy. BC/PP composites [32] have been the focus of research in recent years. For example, Zhang et al. [33,34] used BC particles with different sizes and bamboo powder to modify PP by melt blending. It was shown that BC particles of suitable size were able to greatly enhance the mechanical and other material properties of PP/BC composites.

While BC/PP composites exhibit multifunctional properties, most of the studies to date have only focused on the effect of a small amount of BC on the modification of PP. In addition, little work has been reported on reducing the density of BC/PP composites through foaming. This study aims to further investigate BC/PP composites with the goal of improving their foaming performance and reducing the usage of PP. Ethylene vinyl acetate (EVA) was first used as a foaming modifier for PP. Then, the EVA-modified PP (E-PP) was mixed with BC and foaming agent and other additives. Subsequently, the foamed E-PP/BC composites were prepared using hot-pressing, and the adsorption and mechanical properties were finally evaluated and discussed.

## 2. Results and Discussion

### 2.1. Formation Mechanism of Foamed E-PP/BC Composites

The reason for choosing EVA as a modifier is that EVA is a low-cost and low-density elastomer that can facilitate micro-foaming in the PP matrix. EVA has a high melting point and is able to increase the melt strength of PP, while being conducive to evenly dispersing the gas and to locking the bubbles during the foaming process. In our previous study, the foaming properties of PP and E-PP were compared. The results indicated that the foamed PP had nonuniform and large-sized bubble pores, while the foamed E-PP exhibited more uniform and smaller-sized bubble pores at 5% of EVA [35]. As shown in Figure 1, first, PP was modified by EVA to obtain E-PP. Then, to facilitate dispersion, the additives, comprising AC (foaming agent), ZnO, and Znst, were mixed with E-PP and made into the E-PP/auxiliaries masterbatch using a twin-screw extruder. Subsequently, the BC and E-PP/auxiliaries masterbatches were melt blended to prepare the E-PP/BC/auxiliaries masterbatch, followed by hot-pressing. At this moment, AC was decomposed into masses of nitrogen (N_2_) and CO under high temperatures, and both ZnO and Znst played a role in accelerating thermal decomposition. This is because the N-C, which has a lower bond energy than N=N, is easier to break in order to generate acylamino (
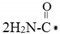
) and N_2_, and then the generated acylaminos react with each other, generating CO and urea. The N_2_ and CO are able to cause the matrix to produce pores, thus resulting in the foamed E-PP/BC composite [36].

### 2.2. Surface Morphologies

SEM images of the foamed E-PP/BC composites with different BC content are shown in Figure 2. The pure foamed E-PP shows amounts of closed pores and some parallel and capacitive pores (Figure 2a). At a small amount of added BC (5 wt.%), the foaming effect is not very different, but the parallel and capacitive pores are reduced (Figure 2b). The results of the mercury injection test are shown in Figure 3. The foamed E-PP/BC (5 wt.%) composite shows an average pore size of about 44.64 μm and a porosity of 24.27%. At an amount of added BC of 20 wt.%, the pores of the foamed composite become irregular (Figure 2c), which might be caused by the difficulty of gas encapsulation due to the melt fluidity of E-PP being reduced with the increase of BC. At this time, its average pore size and porosity is improved to about 34.35 μm and 28.24%, respectively. At amounts of added BC of 30 wt.% and 40 wt.%, the foamed composites show the same irregular pore shape and smaller (10 to 28μm) pore size (Figure 2d,e). As can be seen from the high magnification image in Figure 2e, BC particles were mostly encased, and appeared be poorly bonded by E-PP, resulting in lower mechanical strength. At an added BC amount of 50 wt.%, the pore size and shape of the foamed composite were very uneven, showing poor foaming quality (Figure 2f), resulting from the large amount of BC blocking the flow of E-PP. As such, the average pore size increased to 67 μm, and the porosity decreased to 20.16% (Figure 3). Overall, the foaming quality of the composites seemed to suffer the least with up to 20% addition of BC.

### 2.3. Chemical Component Analysis

To verify the combination mode between BC and E-PP in the composite, BC, E-PP, and their composite were analyzed by FTIR. As shown in Figure 4, the absorption peaks of BC are mainly at 3455 cm^−1^ (–OH), 1590 cm^−1^ (C=C), and 1078 cm^−1^ (C–O). In addition, the absorption peaks of E-PP are mainly at 2953 cm^−1^, 2845 cm^−1^, 1459 cm^−1^, 1377 cm^−1^, 1156 cm^−1^, 1156 cm^−1^, and 971 cm^−1^, and are generated by the symmetric and asymmetric stretching vibrations of –CH_3_ and –CH_2_. However, the foamed E-PP/BC composite does not show new absorption peaks that are different from those of BC and E-PP. Therefore, the combination of BC and PP should be physical.

To check the change of the crystal structure during the preparation process, XRD was analyzed, as shown in Figure 5. BC has two relatively obvious diffraction peaks near 24° and 43°, corresponding to the characteristic diffraction peaks of graphite microcrystalline (001) and (002), respectively, thus confirming that BC is an amorphous carbon structure based on a graphite-like microcrystalline structure, because PP is a substance that is easy to crystallize, and E-PP shows diffraction peaks near 14°, 17°, and 19°, which correspond to the α-crystal PP(110), (040), and (130) planes, respectively. Compared with BC and E-PP, there are no more diffraction peaks in the foamed E-PP/BC composites, thus indicating that their crystal phase does not undergo a change during the preparation process. However, the diffraction peaks become weaker with increasing content of BC. This may be due to the fact that BC as a filler restricts the movement of E-PP molecular chain, thus reducing the interface free energy required for crystallization nucleation.

### 2.4. Density of Foamed E-PP/BC Composites

The change of the density of the foamed composites with different BC content is shown in Figure 6. The density of the pure foamed E-PP was 0.67 g/cm^3^. In addition, the density is enhanced with increasing BC content in the foamed composites. Compared with the composite without BC, the density of the composite (5 wt.%) was only increased to 0.68 g/cm^3^, and the pore size and distribution were not significantly effected (as shown in Figure 2a,b). In contrast, a small amount of BC was able to act as a heterogeneous nucleation agent to provide effective nucleation sites for the foaming agent and to make the pores uniform, reducing the number of parallel and capacitive pores, as well as pore collapse. With increasing BC content, the density of the composite continued to increase, reaching 0.90 g/cm^3^ at a BC content of 40 wt.%, and 0.99 g/cm^3^ at a BC content of 50%. This is because BC has a higher density (as shown in Section 3.1) than E-PP, and the effect of foaming is also affected by it.

### 2.5. Formaldehyde Adsorption Performance

After bamboo pyrolysis, the tubular bundles and parenchyma cells became thinner and their cell cavities became larger, forming the porous structure of BC. On the basis of their different sizes, the pores of BC can generally be divided into three categories: macropores (>50 nm), mesopores (20–50 nm), and micropores (<20 nm), of which, the macropore and mesopores are predominant [37]. The abundance of pores causes the excellent adsorption ability, and this is often used to adsorb harmful gases such as benzene and formaldehyde in indoor air. The test for formaldehyde adsorption is shown in Figure 7a; samples with a size of 10 mm × 10 mm × 4 mm were put into a formaldehyde adsorption testing desiccator at 26 ± 2 °C for 24 h. Figure 7b shows the relationship between the foamed E-PP/BC composites and the formaldehyde adsorption rate. When the amount of BC added was less than 5 wt.%, the composites showed no formaldehyde adsorption ability, because it is easy for a small amount of BC to be completely wrapped by E-PP without the pores being exposed. With increasing BC content, the formaldehyde adsorption rate exhibits a proportional improvement, with the formaldehyde adsorption rate being 0.48 when the amount of BC added was 40 wt.%, and ranging up to 0.6 when the BC content was 50 wt.%. This is because, with increasing BC content, some part of the BC is not covered, leading to a certain adsorption effect.

### 2.6. Mechanical Performance

Static bending strength was tested using a microcomputer-controlled electronic universal testing machine (as shown in Figure 8a), and was generally used to study the ability of materials to bear external loading without failure and also to indirectly characterize the dispersion situation of the fillers in the matrix. As shown in Figure 8b, with increasing BC content, the bending strength of the foamed E-PP/BC composites first goes up and then goes down. At a BC additive amount of 5 wt.%, the bending strength of the foamed composite reaches a maximum of 15.01 MPa, which is 1.6% higher than that of the pure foamed E-PP. This is because at low BC content, the E-PP matrix in the composites plays a leading role, and BC scatters within it, destroying the original single phase, leading to a mechanical interlock with E-PP, thus enhancing their mechanical properties. However, when the BC is increased continuously to a content of 20 wt.%–40 wt.%, their bending strength is actually reduced, although not obviously, reaching 12.9 MPa, 12.45 MPa, and 11.59 MPa, respectively. However, at a BC content of 50 wt.%, the bending strength drops significantly to 8.85 MPa. This is because a large amount of BC causes the contact area between E-PP and BC to decrease, leading to aggregation of BC particles with poor bonding. Moreover, the heterogeneous pores in the foamed E-PP/BC composite are also one of the reasons for poor bending strength. Overall, the foamed E-PP/BC (40 wt.%) composite seems to be the optimal level, because its bending strength is not reduced to the same extent, and meets the Chinese GB/T 9431-2008 standard. In addition, at this level, the formaldehyde adsorption index remains high, being only slightly lower than that at a BC content of 50 wt.%.

## 3. Materials and Methods

### 3.1. Materials

Bamboo Charcoal (BC) particles (average diameter of 750 μm, density of 0.98–1.29 g/cm^3^) were purchased from Jiangshan Lvyi Bamboo Charcoal Co., Ltd., Quzhou, Zhejiang, China; Polypropylene (PP, K8303), density of 0.89–0.91 g/cm^3^ and melt flow rate of 1.0–3.0 g/10 min, was purchased from Suzhou aowigi new material Co., Ltd., Suzhou, China; Azodicarbonamide (AC, A.R.) as a foaming agent (foaming temperature of 195–210 °C), was purchased from Dongguan Haise Plastic Materials Co., Ltd.(Dongguan, Guangdong, China); ZnO and Znst as antioxidants were purchased from BASF AG (Ludwigshafen, Germany); Ethylene vinyl acetate (EVA, VA content of 18%, MI = 2.5), as a modifier, was purchased from Macklin Inc. (Pudong, Shanghai, China). All chemicals were used directly without further modification.

### 3.2. EVA-Modified PP (E-PP)

According to the mass ratio of 95:5, PP and EVA were mixed in the torque rheometer for 10 min at a temperature of 175 °C and a speed of 55 r/min. Subsequently, AC, ZnO, and Znst were added to the above PP/EVA mixture, where their mass was 1.5 wt.%, 0.7 wt.%, and 0.3 wt.%, respectively, followed by being kept blending for 5 min and then drying; after this, the EVA-modified PP (E-PP) was obtained.

### 3.3. Preparation of Foamed E-PP/BC Composites

The foamed E-PP/BC composites were prepared using E-PP as a matrix and BC as fillers. To prepare the E-PP masterbatch well, E-PP was first mixed with foaming agent (AC) and its auxiliaries (ZnO and Znst) by stirring for 5 min in a mixer with a mass rate of 15:7:3, and then moved into a twin-screw extruder, where the first temperature zone of the twin screw extruder was set at 168 °C, the second temperature zone at 173 °C, the third temperature zone at 178 °C, and the rotating speed at 45 r/min. Subsequently, the E-PP masterbatch was further mixed with BC in mass rates according to Table 1 and stirred for 10 min to obtain the E-PP/BC mixture, followed by transferring back to the twin-screw extruder and being smelted for 3 min at the same temperature as the preparation of the E-PP masterbatch, at a rotating speed of 40 r/min. The mixed samples were cut into debris with 10 mm using scissors and transferred to the mold, followed by pressing into a size of 10 mm × 10 mm × 3 mm. Finally, the above samples were put into the customized mold and pressed at 210 °C and 10 MPa for 12 min to obtain the foamed E-PP/BC composites.

### 3.4. Performance Test

#### 3.4.1. Mercury Injection Performance Test

The pore size and porosity of the foamed E-PP/BC composites were determined using an autopore V 9600 automatic mercury injection device. The samples were placed in the oven at 90 °C to dry for 4–5 h before testing. The porosity (K) was calculated according to the following formula: *K* = (1 − ρ_b_/ρ_a_) × 100%, where ρ_b_ and ρ_a_ stand for volume density and apparent density, respectively.

#### 3.4.2. Density Test

As a porous material, BC could absorb moisture from air. Therefore, the method for the measurement of the real density of composite materials was conducted by keeping the composite materials in a drying box until they were completely dried, and then moving them into ethanol of 95% until they reached saturation. The density (ρ) of the composite materials was calculated using the formula ρ = *mc*/(*ml*/ρ*l*), where *mc* stands for the absolute dry mass, *ml* for the mass after absorption-saturation, and ρ*l* for the density after absorption-saturation.

#### 3.4.3. Formaldehyde Adsorption Performance Test

The samples were cut into a size of 10 mm × 10 mm × 4 mm. Before testing, the samples were placed in an oven at 80 °C for 24 h, and the samples were weighed after drying. Subsequently, the samples were put into a formaldehyde adsorption testing desiccator at 26 ± 2 °C to test the mass after formaldehyde adsorption for 24 h until the weight remained unchanged. The formaldehyde adsorption rate of the samples was calculated according to the equation *AJ* = (*mt* − *mo*)/*mo* × 100%, where *AJ*, *mo*, and *mt* stand for formaldehyde adsorption rate, and the mass before and after immersion, respectively.

#### 3.4.4. Mechanical Properties Test

According to the Chinese GB/T 9431-2008 standard, an electronic universal testing machine was used to test the static bending strength of the samples. The materials were made into a standard specimen with a size of 80 mm × 10 mm × 4.5 mm. The test span was set at 70 mm, and the tensile rate was 10 mm/min. The results were estimated based on the formula σ_1_ = 3*FL*/(2*bh*^2^), where σ_1_ stands for bending strength, *F* for the external force, *L* for span, *b* for width of specimen, and *h* for thickness of specimen.

### 3.5. Characterization

Fourier infrared spectrometer (FTIR, Perkin Elmer, Akron, OH, USA) was used to characterize the chemical structures of the BC, E-PP, and E-PP/BC samples in the wave number range of 400–4000 cm^−1^ by scanning 32 times. The FTIR samples were prepared by mixing the three kinds of powder with KBr at a mass rate of 1:100, followed by pressing into a circular wafer. X-ray Diffraction (XRD-6000, Shimadzu, Tokyo, Japan) was used to analyze the crystallinity of the samples with a scanning range of 2°–80° (2θ) and scanning speed of 2°/min, and the operating voltage was set at 40 KV, the tube current at 30 mA, and the wavelength at 0.15406 nm. Scanning electron microscopy (SEM, Hitachi SU 8010, Tokyo, Japan) was used to observe surface morphologies of different samples at the emission voltage of 15 KV. The composites were soaked in liquid nitrogen for 30 s to quench, and then the quenched composites were cut into a thickness of 2 mm and sprayed with gold to get the SEM samples.

## 4. Conclusions

Foamed EVA-modified PP/BC composites were successfully prepared with up to 50% of BC as replacement for PP. The BC content dictates the properties of the foamed E-PP/BC composites. At a low BC content (5 wt.%), the foamed E-PP/BC composites exhibit no obvious change in their pore size, density, formaldehyde adsorption, or bending strength. However, with increasing content of BC, the foamed E-PP/BC composites exhibit a significant improvement in density and formaldehyde adsorption, but a steady decrease in the bending strength. With all factors considered, the foamed E-PP/BC (40 wt.%) composite achieved the optimal performance, with a density of 0.90 g/cm^3^, 24-h formaldehyde adsorption rate of 0.48, and bending strength of 11.59 MPa. Low density, and good strength and adsorptive properties make the foamed E-PP/BC composites suitable for many interior paneling applications.

## Figures and Tables

**Figure 1 materials-14-01524-f001:**
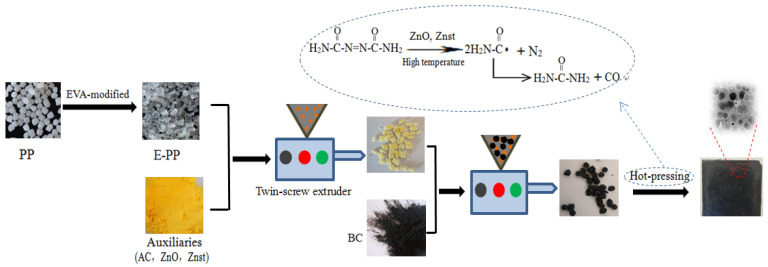
Preparation process and forming mechanism of the foamed E-PP/BC composites.

**Figure 2 materials-14-01524-f002:**
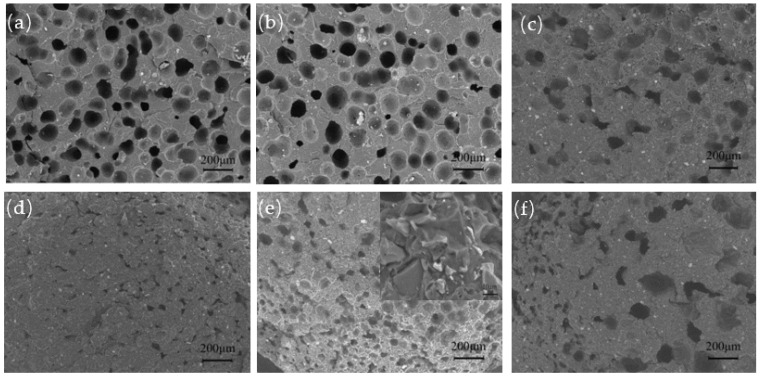
SEM images of the foamed E-PP/BC composites with BC content of (**a**) 0 wt.%, (**b**) 5 wt.%, (**c**)20 wt.%, (**d**) 30 wt.%, (**e**) 40 wt.%, and (**f**) 50 wt.%.

**Figure 3 materials-14-01524-f003:**
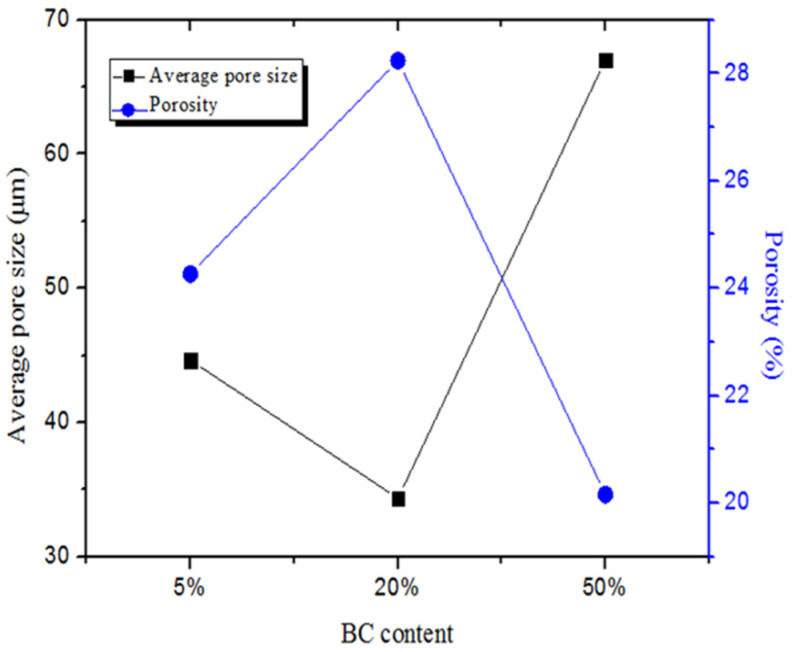
The average pore size and the porosity of the foamed E-PP/BC composites with different BC content.

**Figure 4 materials-14-01524-f004:**
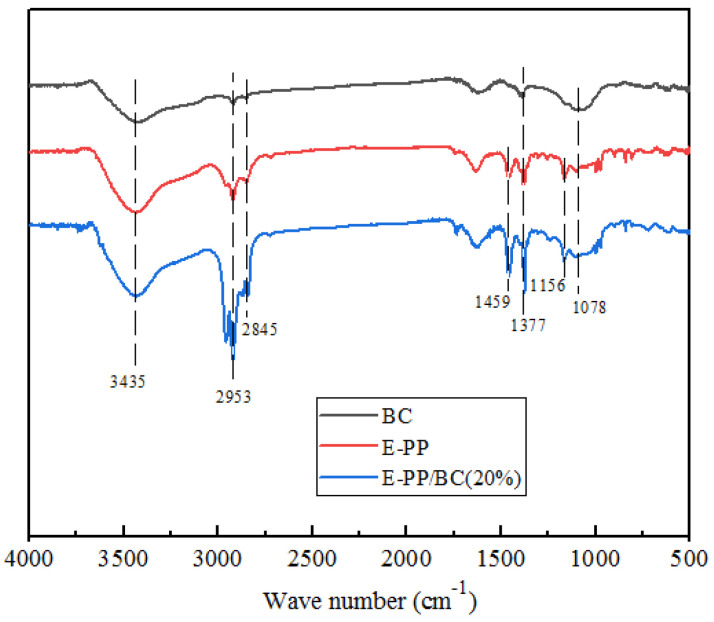
FTIR spectra of BC, E-PP, and the E-PP/BC (20 wt.%) composite.

**Figure 5 materials-14-01524-f005:**
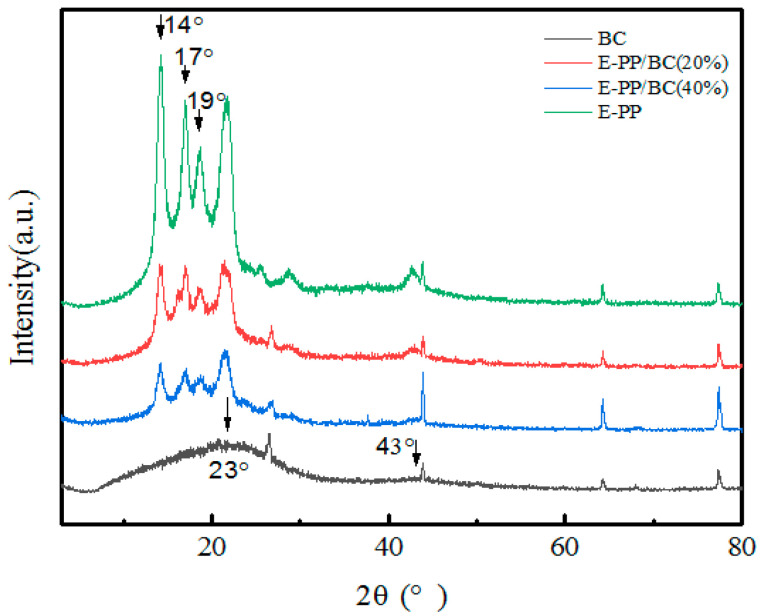
XRD patterns of BC, E-PP, and foamed E-PP/BC (20 wt.% and 40 wt.%) composites.

**Figure 6 materials-14-01524-f006:**
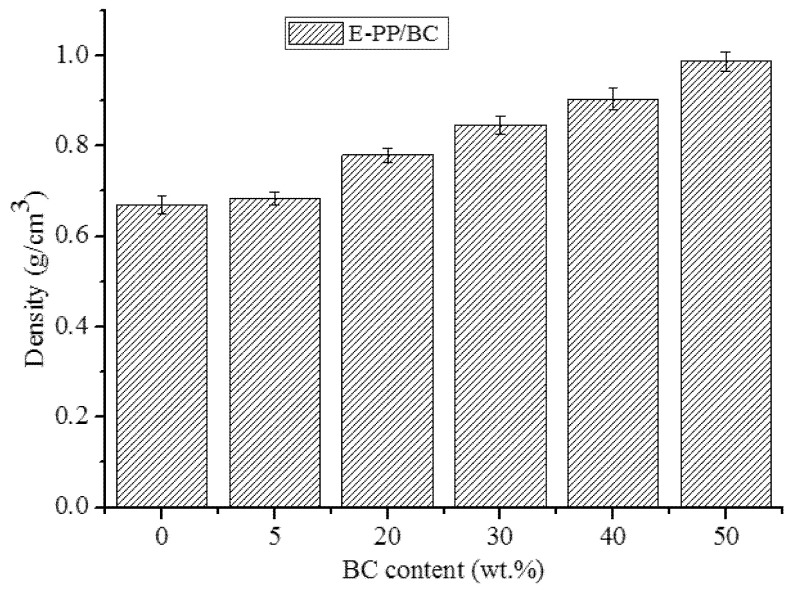
Change in the density of the foamed E-PP/BC composites with different BC content.

**Figure 7 materials-14-01524-f007:**
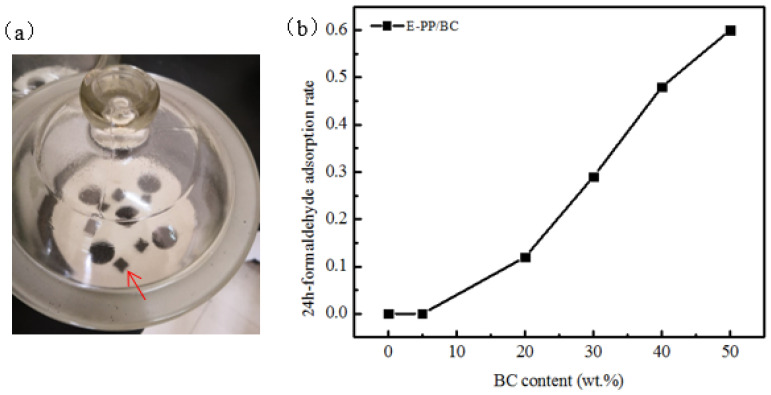
Formaldehyde adsorption testing and results: (**a**) formaldehyde adsorption testing desiccator, and (**b**) 24-h formaldehyde adsorption rate curves of the foamed E-PP/BC composites with different BC content.

**Figure 8 materials-14-01524-f008:**
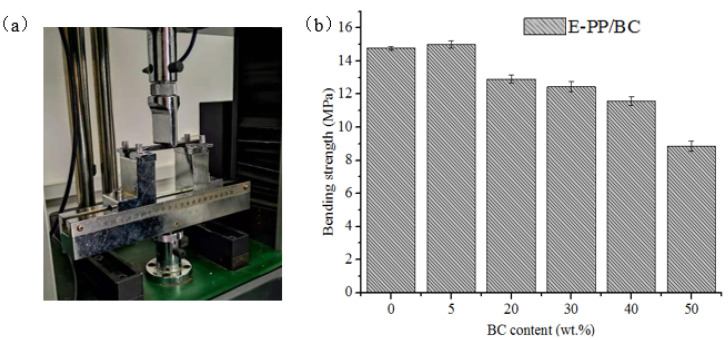
Mechanical testing and results: (**a**) picture of static mechanical performance testing and (**b**) bending strength of foamed E-PP/BC composites with different BC content.

**Table 1 materials-14-01524-t001:** The mixing component content of foamed E-PP/BC composites.

No.	BC/wt.%	E-PP/wt.%	Foaming Agent and its Auxiliaries/wt.% (AC, ZnO, Znst)
1	0	97.5	2.5
2	5	92.5	2.5
3	20	77.5	2.5
4	30	67.5	2.5
5	40	57.5	2.5
6	50	47.5	2.5

## Data Availability

The data presented in this study are available on request from the corresponding author.

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
