# Peer review of "Mechanical and Adsorptive Properties of Foamed EVA-Modified Polypropylene/Bamboo Charcoal Composites"

_materials, 2021, doi:10.3390/ma14061524_

Round 1
Reviewer 1 Report
After reading the manuscript entitled "Research on preparation and mechanical and adsorptive properties of EVA-modified polypropylene / bamboo charcoal composite foam" by
Wenzhu Li, Jing Zhang, Jingda Huang, Yuanchao Shao, Wenbiao Zhang and Chunping Dai I believe that the work is interesting and prepared correctly according to the art of scientific work. However, it requires a little enrichment.
My questions and doubts:
I suggest moving Chapter 3 (p. 6 MAterials and methods) before discussing the results. Please correct the chapter numbering.
The notation PMa is used incorrectly, also in Fig. 7.
The BC additives density results are missing.
Did the PP use exhibit an amorphous structure? Is it related to its structure or to the composite manufacturing process? PP is a material that crystallizes easily, therefore I was interested in its amorphic structure revealed in the XRD results. Here, DSC tests should be performed to estimate the influence of fillers on the crystallinity of the matrix.
What is the reaction temperature of the blowing agent? Could the material have been foamed already at the stage of masterbatch production?
Author Response
After reading the manuscript entitled "Research on preparation and mechanical and adsorptive properties of EVA-modified polypropylene / bamboo charcoal composite foam" by
Wenzhu Li, Jing Zhang, Jingda Huang, Yuanchao Shao, Wenbiao Zhang and Chunping Dai I believe that the work is interesting and prepared correctly according to the art of scientific work. However, it requires a little enrichment.
My questions and doubts:
- I suggest moving Chapter 3 (p. 6 MAterials and methods) before discussing the results. Please correct the chapter numbering.
- Response: Thanks for the good advice. But the “MAterials and methods” following the “Discussion” is the requirement for the journal typesetting.
- The notation PMa is used incorrectly, also in Fig. 7.Figure 8. (b)bending strength of foamed E-PP/BC composites with different BC content.
- Response: Thanks for the good advice. The error has been modified in the revised manuscript.
- The BC additives density results are missing.
- Response: Thanks for the good advice. The air-dried density of the BC additives is 0.98~1.29g/cm³ and added to “Materials” section of the revised manuscript.
- Did the PP used exhibit an amorphous structure? Is it related to its structure or to the composite manufacturing process? PP is a material that crystallizes easily, therefore I was interested in its amorphic structure revealed in the XRD results. Here, DSC tests should be performed to estimate the influence of fillers on the crystallinity of the matrix.
- Response: Thanks for the good advice. The XRD pattern of the samples has been remeasured. As shown in Fig.5 of the revised manuscript.
Figure 5. XRD patterns of BC, E-PP, and foamed E-PP/BC (20 and 40wt.%) composites.
To check the change of their crystal structure in the preparation process, the XRD was analyzed and shown in Fig. 5. BC has two relatively obvious diffraction peaks near 24° and 43° corresponding to the characteristic diffraction peaks of graphite microcrystalline (001) and (002), respectively, which confirms that BC is an amorphous carbon structure based on graphite-like microcrystalline. Because PP is a easy to crystallize substance, and E-PP shows the diffraction peaks near 14°, 17°, and 19°, respectively, corresponding to the α-crystal PP(110), (040), and (130) plane. Compared with BC and E-PP, there are no more diffraction peaks in the foamed E-PP/BC composties, which indicates that their crystal phase does not have a change during the preparation process. However, the diffraction peaks become weak with the increasing content of BC. This may be due to the fact that BC as a filler restricts the movement of E-PP molecular chain, thus reducing the interface free energy required for crystallization nucleation.
- What is the reaction temperature of the blowing agent? Could the material have been foamed already at the stage of masterbatch production?
- Response: Thanks for the good advice. The reaction temperature of the blowing agent is 195~210℃ and has been added to the “Materials” section of the revised manuscript. The materials have been foamed at the stage of hot-pressing (as shown in Fig.1), not at the stage of masterbatch production.

Reviewer 2 Report
The manuscript of W. Li et al. deals with EVA-modified polypropylene/bamboo charcoal composite foam obtained using a twin-screw extruder, following by hot-pressing at high temperature. Specifically, the effect of filler content on the composite's porosity, absorption, and mechanical properties has been investigated.
The research is relevant to the material journal.
However, some points in the manuscript need to be revised, some experiment information is missing, and some parts of the manuscript need punctual clarifications. For this, the following MAJOR REVISIONS are suggested. Please find detailed notes on the pdf file attached. Hereafter a summary of the main comments:
- The Materials and process section should be moved before the results;
- Further details on SEM sample preparation should be added
- In paragraph 2.2. Surface morphologies. To improve the results' readability, a table or graph could be used, reporting the pores' median dimension and the porosity amount wrt the BC loading.
- Line 133 the XRD peaks are diffraction peaks and not absorption.
- Paragraph 3.3 figure 4 Could please the authors clarify the E-PP crystallization mechanism? Could be the peaks present in the E-PP/BC composite just due to the BC presence (which present some diffraction peaks itself) and not to the E-PP crystal? E-PP is amorphous and has no obvious diffraction peak. Could be useful to provide a DSC analysis of the E-PP and E-PP/BC composite to analyze the crystallinity if there is any.
- Please check the manuscript for English and typing errors
Author Response
The manuscript of W. Li et al. deals with EVA-modified polypropylene/bamboo charcoal composite foam obtained using a twin-screw extruder, following by hot-pressing at high temperature. Specifically, the effect of filler content on the composite's porosity, absorption, and mechanical properties has been investigated.
The research is relevant to the material journal.
However, some points in the manuscript need to be revised, some experiment information is missing, and some parts of the manuscript need punctual clarifications. For this, the following MAJOR REVISIONS are suggested. Please find detailed notes on the pdf file attached. Hereafter a summary of the main comments:
- The Materials and process section should be moved before the results;
Response: Thanks for the good advice. But the “Materials and methods” following the “Discussion” is the requirement for the journal typesetting.
- Further details on SEM sample preparation should be added
Response: Thanks for the good advice. The details on SEM sample preparation have been added to Characterization section of the revised manuscript. As following:
The composites were soaked in liquid nitrogen for 30s to quench, and then the quenched composites were cut into a thickness of 2 mm and sprayed with gold to get the SEM samples.
- In paragraph 2.2. Surface morphologies. To improve the results' readability, a table or graph could be used, reporting the pores' median dimension and the porosity amount wrt the BC loading.
Response: Thanks for the good advice. We have made a graph (as following) about the pores' median dimension and the porosity as shown in Fig. 3 of the revised manuscript.
Figure 3. The average pore size and the porosity of the foamed E-PP/BC composites with different BC content.
- Line 133 the XRD peaks are diffraction peaks and not absorption.
Response: Thanks for the good advice. The error has been corrected. As follow,
corresponding to the characteristic diffraction peaks of graphite microcrystalline (001) and (002).
- Paragraph 3.3 figure 4 Could please the authors clarify the E-PP crystallization mechanism? Could be the peaks present in the E-PP/BC composite just due to the BC presence (which present some diffraction peaks itself) and not to the E-PP crystal? E-PP is amorphous and has no obvious diffraction peak. Could be useful to provide a DSC analysis of the E-PP and E-PP/BC composite to analyze the crystallinity if there is any.
Response: Thanks for the good advice. To further verify, the XRD pattern of the samples has been remeasured. As shown in Fig.5a-b of the revised manuscript.
Figure 5. XRD patterns of BC, E-PP, and foamed E-PP/BC (20 and 40wt.%) composites.
To check the change of their crystal structure in the preparation process, the XRD was analyzed and shown in Fig. 5. BC has two relatively obvious diffraction peaks near 24° and 43° corresponding to the characteristic diffraction peaks of graphite microcrystalline (001) and (002), respectively, which confirms that BC is an amorphous carbon structure based on graphite-like microcrystalline. Because PP is a easy to crystallize substance, and E-PP shows the diffraction peaks near 14°, 17°, and 19°, respectively, corresponding to the α-crystal PP(110), (040), and (130) plane. Compared with BC and E-PP, there are no more diffraction peaks in the foamed E-PP/BC composties, which indicates that their crystal phase does not have a change during the preparation process. However, the diffraction peaks become weak with the increasing content of BC. This may be due to the fact that BC as a filler restricts the movement of E-PP molecular chain, thus reducing the interface free energy required for crystallization nucleation.
- Please check the manuscript for English and typing errors
Response: Thanks for the good advice. We have done our best to check and perfect the language problem and typing errors, and also asked native English speakers for polishing it. The modification could be seen in the whole paper.
Reviewer 3 Report
There are some weaknesses through the manuscript which need improvement. Therefore, the submitted manuscript cannot be accepted for publication in this form, but it has a chance of acceptance after a major revision. My comments and suggestions are as follows:
1- Abstract gives information on the main feature of the performed study, but some details about the conducted tests must be added.
2- It would be nice, if authors can use a short title for manuscript.
3- Authors must clarify necessity of the performed research. Aims and objectives of the study, and also differences with the previous researches must be clearly mentioned in the last part of introduction.
4- The literature study must be enriched. In this respect, authors must read and refer to the following papers: (a) https://doi.org/10.4028/www.scientific.net/AMM.110-116.1361 (b) https://doi.org/10.1016/j.matpr.2020.08.792
5- It would be nice, if authors could add figures of fabricated composite in the first section of section, and then illustrate SEM images.
6- As it is an experimental study, authors must present some figures in different sub-sections (e.g., 3.4 and 3.6, and 2.3.1) to show specimens under test conditions.
7- Why this particular BC contents are considered?
8- Details of experimental test with at least a figure (experimental setup) must be presented in 3.6. It is necessary.
9- In 2.3.4 experiment must be explained in details (e.g., setup. formula, results, calculation and so on).
10- In its language layer, the manuscript should be considered for English language editing. There are sentences which have to be rewritten.
11- The conclusion must be more than just a summary of the manuscript. List of references must be updated based on the proposed papers. Please provide all changes by red color in the revised version.
Author Response
There are some weaknesses through the manuscript which need improvement. Therefore, the submitted manuscript cannot be accepted for publication in this form, but it has a chance of acceptance after a major revision. My comments and suggestions are as follows:
1- Abstract gives information on the main feature of the performed study, but some details about the conducted tests must be added.
Response: Thanks for the good advice. The Abstract has been rewritten and some details about the conducted tests have been added. As following:
Due to its excellent adsorption and humidity control function, bamboo charcoal (BC) has often been used to mix with polypropylene (PP) to produce PP/BC composites for interior paneling applications. However, due to poor foaming quality of PP, PP/BC composites are suffering from high density which limits its scope of use. Here, to improve its foaming quality, PP was modified with ethylene vinyl acetate (EVA), and then the EVA-modified PP (E-PP) was mixed with BC with different contents (0~50wt.%), foaming agent (Azodicarbonamide, AC) and its auxiliaries (ZnO, Znst) in a twin-screw extruder, followed by hot-pressing at high temperature to obtain foamed E-PP/BC composites. The resulting composites showed good porosity and pore distribution with the BC content increasing up to 20%. Further increase in the BC content seemed to cause the foaming performance to decrease significantly. The product density and adsorption rate increased, while the mechanical strength decreased with increasing BC content. At the BC content of 40wt.%, the foamed E-PP/BC composite shows the best combined performance with density of 0.90g/cm3, 24h-formaldehyde adsorption rate of 0.48, and bending strength of 11.59MPa.
2- It would be nice, if authors can use a short title for manuscript.
Response: Thanks for the good advice. We have used a shorter title. As follow,
Mechanical and adsorptive properties of foamed EVA-modified polypropylene/bamboo charcoal composites
3- Authors must clarify necessity of the performed research. Aims and objectives of the study, and also differences with the previous researches must be clearly mentioned in the last part of introduction.
Response: Thanks for the good advice. The introduction has been rewritten and aims, objectives and differences of the study has been mentioned. As following:
While BC/PP composites show multifunctional properties, most of the studies to date have only focused on the effect of a small amount of BC on the modification of PP. In addition, little work has been reported about reducing the density of BC/PP composites by foaming. This study aims at further investigating BC/PP composites with a goal to improve the foaming performance and reduce the usage of PP. Ethylene vinyl acetate (EVA) was first used as the foaming modifier for PP. Then, the EVA-modified PP (E-PP) was mixed with BC and foaming agent and other additives. Subsequently, the foamed E-PP/BC composite were prepared using hot-pressing and finally the adsorption and mechanical properties were evaluated and discussed.
4- The literature study must be enriched. In this respect, authors must read and refer to the following papers: (a) https://doi.org/10.4028/www.scientific.net/AMM.110-116.1361 (b) https://doi.org/10.1016/j.matpr.2020.08.792
Response: Thanks for the good advice. We have red and cited the two literatures as references [7-8]. As follow,
[7]Diaz-Quijada, G. A.; Peytavi, R.; Nante, A.; et al. Surface modification of thermoplastics—towards the plastic biochip for high throughput screening devices[J]. Lab on a Chip, 2007, 7, 856-862.
[8]Fabiyi, J. S.; McDonald, A. G.;, McIlroy, D. Wood modification effects on weathering of HDPE-based wood plastic composites[J]. Journal of Polymers and the Environment, 2009, 17, 34-48.
5- It would be nice, if authors could add figures of fabricated composite in the first section of section, and then illustrate SEM images.
Response: Thanks for the good advice. We have added some figures to the first section of Fig.1. As following:
Figure 1. Preparation process and forming mechanism of the foamed E-PP/BC composites.
We agree with that it would be nice that the SEM images of every stage in the fabrication process of the composites were given, but the amounts of the current SEM images should be also acceptable and could meet the readers’ understanding. Thank you .
6- As it is an experimental study, authors must present some figures in different sub-sections (e.g., 3.4 and 3.6, and 2.3.1) to show specimens under test conditions.
Response: Thanks for the good advice. We agree with that it would be very nice to show specimens under test conditions. But some tests were commissioned, therefore, there are no pictures left. And the saved ones were added to Figures 7 and 8 of the revised manuscript. As following,
Figure 7.(a)formaldehyde adsorption testing desiccator.
Figure 8.(a)Picture of static mechanical performance testing.
7- Why this particular BC contents are considered?
Response: Thanks for the good advice. The considered BC contents are setuped according to our preliminary study and there is better research value in this range. Especially when the BC content exceeds 50wt.%, the composite shows very low strength.
8- Details of experimental test with at least a figure (experimental setup) must be presented in 3.6. It is necessary.
Response: Thanks for the good advice. We have added a figure about experimental setup to 3.6 section (Fig.8a) (2.6 section of the revised manuscript ). As following:
Figure 8. (a) Picture of static mechanical properties test
Details of experimental test has been described at details in the “2.3.4 Mechanical properties test” section of the revised manuscript.
9- In 2.3.4 experiment must be explained in details (e.g., setup. formula, results, calculation and so on).
Response: Thanks for the good advice. We have added a detailed explanation in t “2.3.4 Mechanical properties test” section of the revised manuscript. As following ,
3.4.4 Mechanical properties test
According to Chinese GB/T 9431-2008 standard, an electronic universal testing machine was used to test the static bending strength of the samples. The materials were made into a standard specimen with a size of 80×10×4.5 mm. The test span was set as 70 mm, and the tensile rate was 10 mm/min. The result could be estimated based on this formula σ1=3FL/(2bh2), and σ1 stands for bending strength, F for the external force, L for span, b for width of specimen, and h for thickness of specimen.
10- In its language layer, the manuscript should be considered for English language editing. There are sentences which have to be rewritten.
Response: Thanks for the good advice. We have done our best to check and modified the language, and asked native English speakers for polishing it. The modification could be seen in the whole revised manuscript.
11- The conclusion must be more than just a summary of the manuscript. List of references must be updated based on the proposed papers. Please provide all changes by red color in the revised version.
Response: Thanks for the good advice.The conclusion has been rewritten. As following:
Foamed EVA-modified PP/BC composites have been successfully prepared with up to 50% of BC as replacement for PP. The BC content dictates the properties of the foamed E-PP/BC composites. At the low BC content (5wt%), the foamed E-PP/BC composites have no obvious change on their pore size, density, formaldehyde adsorption, and bending strength. However, with the increasing content of BC, the foamed E-PP/BC composites have a significant improvement in the density and formaldehyde adsorption, but decrease on the bending strength. With all factors considered, the foamed E-PP/BC (40 wt.%) composite reaches the optimal performance with the density of 0.90g/cm3, 24h-formaldehyde adsorption rate of 0.48, and bending strength of 11.59MPa. Light density, good strength and adsorptive properties could allow the foamed E-PP/BC composites to be used in replacement of pure plastics for many interior paneling applications.
The literatures have been cited as references[7-8].

Reviewer 4 Report
In the manuscript by Wenzhu Lia et al., the authors present research on preparation and characterization of EVA-modified polypropylene/bamboo charcoal composite foams. In my opinion, the paper is publishable in Materials; however, after major revision. The authors should address the following issues before this work can be accepted for publication.
1. Page 1, Keywords. Please add “formaldehyde”.
2. Pages 1 and 2, Introduction. The literature on formaldehyde adsorption is very extensive. Unfortunately, the authors omitted some important papers, for example, 10.1016/j.carbon.2018.05.023, 10.14710/ijred.1.3.75-80, doi.org/10.1021/es104286d.
3. Page 4, Fig. 3. The legend should be lowered and it should be inside the drawing panel - there is enough space to fit it. The lack of y-axis.
4. Page 4, Fig. 3. A very serious drawback is the lack of data for E-PP/BC with BC content of 50 wt.%!!! This problem also applies to further data collected in Fig. 4.
5. Page 4, Fig. 4. Why are the data for BC sample truncated for higher values of 2theta? It also seems to me that the mutual scale of intensity for BC and E-PP/BC samples is wrong.
6. Page 5, 3.4. What is the density? apparent? mercury? helium? Others?
7. Page 5, Fig. 5. Please add data for BC as solid horizontal line – treated as the reference system.
8. Page 5, Fig. 6. Have the authors studied formaldehyde adsorption for BC?
9. Page 5, lines 167-170. The best way to characterize a porous structure is to perform low-temperature nitrogen adsorption. These measurements are necessary!!!
10. Page 7, line 236. Desorption/Degassing temperature (90oC) is too low - not even water will be removed.
11. Page 7, lines 237 and 238 and Fig. 5. In my opinion it would be nice to show porosity in Fig. 5 as well.
12. Pages 7 and 8, lines 244 and 245, 252 and 253. Problem with subscripts.
13. Page 9, References. Some errors – see ref. 18 and 23. [18]? [23]?
Author Response
In the manuscript by Wenzhu Lia et al., the authors present research on preparation and characterization of EVA-modified polypropylene/bamboo charcoal composite foams. In my opinion, the paper is publishable in Materials; however, after major revision. The authors should address the following issues before this work can be accepted for publication.
- Page 1, Keywords. Please add “formaldehyde”.
Response: Thanks for the good advice. The “Formaldehyde” has been added to Keywords.
- Pages 1 and 2, Introduction. The literature on formaldehyde adsorption is very extensive. Unfortunately, the authors omitted some important papers, for example, 10.1016/j.carbon.2018.05.023, 10.14710/ijred.1.3.75-80, doi.org/10.1021/es104286d.
Response: Thanks for the good advice. The literatures have been cited as references[28-30]. As follow,
[28]Suresh, S.; Bandosz, T. J. Removal of formaldehyde on carbon-based materials: A review of the recent approaches and findings[J]. Carbon, 2018, 137, 207–221.
[29]Carter, E. M.; Katz, L. E.; Speitel, G. E.; et al. Gas-phase formaldehyde adsorption isotherm studies on activated carbon: Correlations of adsorption capacity to surface functional group density[J]. Environmental Science & Technology, 2011, 45, 6498.
[30]Rengga, W.D.P; Sudibandriyo, M.; Nasikin, M. Development of formaldehyde adsorption using modified activated carbon-A review[J]. Int. Journal of Renewable Energy Development, 2012, 1, 75-80.
- Page 4, Fig. 3. The legend should be lowered and it should be inside the drawing panel - there is enough space to fit it. The lack of y-axis.
Response: Thanks for the good advice. We have added a y-axis to Fig. 3 (Fig.4 in the revised manuscript) and the drawing panel has been used. As following:
Figure 4. FTIR spectra of BC, E-PP, and the E-PP/BC ( 20 wt.%) composite.
- Page 4, Fig. 3. A very serious drawback is the lack of data for E-PP/BC with BC content of 50 wt.%!!! This problem also applies to further data collected in Fig. 4.
Response: Thanks for the good advice. In Fig. 3 (Fig.4 in the revised manuscript), We had made a comparison among BC, E-PP, and E-PP/BC by FTIR to see if there is a chemical bond between E-PP and BC. And due to the same preparation conditions, What is the BC content should had no effect on FTIR of the E-PP/BC composites. Thank you.
- Page 4, Fig. 4. Why are the data for BC sample truncated for higher values of 2theta? It also seems to me that the mutual scale of intensity for BC and E-PP/BC samples is wrong.
Response: Thanks for the good advice. To further verify, the XRD pattern of the samples has been remeasured. As shown in Fig.5 of the revised manuscript.
Figure 5. XRD patterns of BC, E-PP, and E-PP/BC composites with BC contents of 20 and 40wt.%.
To check the change of their crystal structure in the preparation process, the XRD was analyzed and shown in Fig. 5. BC has two relatively obvious diffraction peaks near 24° and 43° corresponding to the characteristic diffraction peaks of graphite microcrystalline (001) and (002), respectively, which confirms that BC is an amorphous carbon structure based on graphite-like microcrystalline. Because PP is a easy to crystallize substance, and E-PP shows the diffraction peaks near 14°, 17°, and 19°, respectively, corresponding to the α-crystal PP(110), (040), and (130) plane. Compared with BC and E-PP, there are no more diffraction peaks in the foamed E-PP/BC composties, which indicates that their crystal phase does not have a change during the preparation process. However, the diffraction peaks become weak with the increasing content of BC. This may be due to the fact that BC as a filler restricts the movement of E-PP molecular chain, thus reducing the interface free energy required for crystallization nucleation.
- Page 5, 3.4. What is the density? apparent? mercury? helium? Others?
Response: Thanks for the good advice. In 3.4 (2.4 in the revised manuscript), the headline “Density determination and analysis ” has been changed into “Density of foamed E-PP/BC composites”.
- Page 5, Fig. 5. Please add data for BC as solid horizontal line-treated as the reference system.
Response: Thanks for the good advice. We agree with you so much, but the density of BC is in the range from 0.98 to 1.29g/cm³ supported by the manufacturer. There is no fixed value given, it is difficult to add to Fig. 5 (Fig.6 of the revised manuscript). However, we think it is also acceptable to add the density range to “materials” section and could also as reference system to meet readers’ understanding in the current figure. Thank you.
- Page 5, Fig. 6. Have the authors studied formaldehyde adsorption for BC?
Response: Thanks for the good advice. Here, we mainly study the effect of BC content on mechanical and formaldehyde adsorptive properties of the composites. We agree with very much that it would be nice to have a study on formaldehyde adsorption for BC. In our next article, what we're going to do is that a systematic study on all the adsorption properties of BC and its composites . Thank you.
- Page 5, lines 167-170. The best way to characterize a porous structure is to perform low-temperature nitrogen adsorption. These measurements are necessary!!!
Response: Thanks for the good advice. We also agree with that it is a best way to to perform low-temperature nitrogen adsorption for characterization of a porous structure. In view of that the foamed E-PP/BC composite mainly produced large-size pores by foaming, and nitrogen adsorption is mainly used for mesoporous analysis. But We really agree with you and plan to test it in the next step, thank you.
- Page 7, line 236. Desorption/Degassing temperature (90oC) is too low - not even water will be removed.
Response: Thanks for the good advice. Here's how we think about that too high temperature might affect properties of the samples, so the temperature was used to dry for 24h. But the we are sure that the samples were weighted many times to make sure them dried.
- Page 7, lines 237 and 238 and Fig. 5. In my opinion it would be nice to show porosity in Fig. 5 as well.
Response: Thanks for the good advice. We agree with you very much, and we have given the porosity in Fig.3 of the revised manuscript. As follow,
Figure 3. The average pore size and the porosity of the foamed E-PP/BC composites with different BC content.
- Pages 7 and 8, lines 244 and 245, 252 and 253. Problem with subscripts.
Response: Thanks for the good advice. That has been corrected.
- Page 9, References. Some errors – see ref. 18 and 23. [18]? [23]?
Response: Thanks for the good advice. That has been corrected.

Round 2
Reviewer 1 Report
The authors made changes to the manuscript that increase the scientific value of the work, therefore, in my opinion, the work is suitable for publication in the Materials journal in a present form.
Author Response
Thank you for your recognition to our article.
Reviewer 2 Report
The authors revised their manuscript addressing all the comments. The revised version can be accepted for publication.
Author Response

(The authors gave the same response as above.)

Reviewer 3 Report
Please remove [J] and [D] at the end of in all references in the reference list.
The paper has been improved and corresponding modifications have been conducted. In my opinion, the current version can be considered for publication
Author Response

(The authors gave the same response as above.)

Reviewer 4 Report
The authors revised their manuscript well. The revised version can be accepted for publication. However,
- Fig. 2(f). Problem with “(f)”.
- Fig. 4, legend. E-PP/BC ---> E-PP/BC (205). See Fig. 5.
- Fig. 8(b), legend. Red lines/square?
Author Response
Thank you for your recognition to our article
Comments and Suggestions for Authors
The authors revised their manuscript well. The revised version can be accepted for publication. However,
Fig. 2(f). Problem with “(f)”.
Response: Thanks for the good advice. But it has not been found that what is wrong with (f) in Fig.2(f).
Fig. 4, legend. E-PP/BC ---> E-PP/BC (20 wt.%). See Fig. 5.
Response: Thanks for the good advice. The legend in Fig.4 has been corrected. As following,
Fig. 8(b), legend. Red lines/square?
In addition, the Fig.3 has been modified as following,
Response: Thanks for the good advice. The legend in Fig.8(b) has been corrected. As following,
